# Memristor-based analogue computing for brain-inspired sound localization with in situ training

Bin Gao [1,2 ✉], Ying Zhou[1,2], Qingtian Zhang[1,2], Shuanglin Zhang[1], Peng Yao[1], Yue Xi[1], Qi Liu[1], Meiran Zhao[1], Wenqiang Zhang [1], Zhengwu Liu [1], Xinyi Li[1], Jianshi Tang[1], He Qian[1] & Huaqiang Wu [1 ✉]

The human nervous system senses the physical world in an analogue but efficient way. As a crucial ability of the human brain, sound localization is a representative analogue computing task and often employed in virtual auditory systems. Different from well-demonstrated classification applications, all output neurons in localization tasks contribute to the predicted direction, introducing much higher challenges for hardware demonstration with memristor arrays. In this work, with the proposed multi-threshold-update scheme, we experimentally demonstrate the in-situ learning ability of the sound localization function in a 1K analogue memristor array. The experimental and evaluation results reveal that the scheme improves the training accuracy by ~45.7% compared to the existing method and reduces the energy consumption by ~184× relative to the previous work. This work represents a significant advance towards memristor-based auditory localization system with low energy consumption and high performance.

[1] School of Integrated Circuits, Beijing National Research Center for Information Science and Technology (BNRist), Tsinghua University, 100084 Beijing, China. [2] These authors contributed equally: Bin Gao, Ying Zhou, Qingtian Zhang. ✉email: gaob1@tsinghua.edu.cn; wuhq@tsinghua.edu.cn

The detection of sound sources is a basic function of human beings and can be acquired through multiple training processes[1,2]. In the biological brain, with neurons and synapses, binaural auditory information is processed to localize the sound source. Both the environment and the physiological structure would influence the auditory effect, specifically manifested in the time difference of binaural soundwaves, spectral shape and so on[2–4]. Complementary metal-oxide-semiconductor (CMOS) circuits have been widely employed to detect the received time difference of binaural sound signals[5–9]. As shown in Fig. 1a, most of them relied on the interaural time difference (ITD) theory[10], using the time difference between the sound reaching two ears to localize the direction of the sound source. This simplified model can be implemented on existing CMOS hardware. However, due to the loss of partial auditory information, the angle detection range is certainly limited[3]. In addition, with the explosive growth of analysis data for real-time sound information, due to the separation of the processing unit and memory, the traditional CMOS hardware platform with von Neumann architecture is facing the obstacle that computing efficiency is gradually failing to keep up with the increasing demand[11,12].

Driven by the continuously increasing demand for computing power, the efficient bio-inspired computing has become one of the most promising paradigms for overcoming the computing bottleneck. The emerging electronic devices, memristors, show similarity to biological synapses[13–17]. Such a device is able to play roles in both storage and dealing with information analogously, promoting the idea of computation-in-memory. It has also drawn great attention for its low power consumption and high-density features in neuromorphic applications[18–22]. In many representative classification tasks, the combination of the memristor and bio-inspired architecture has demonstrated its superiority in high parallelism and energy efficiency[23–34]. However, these tasks

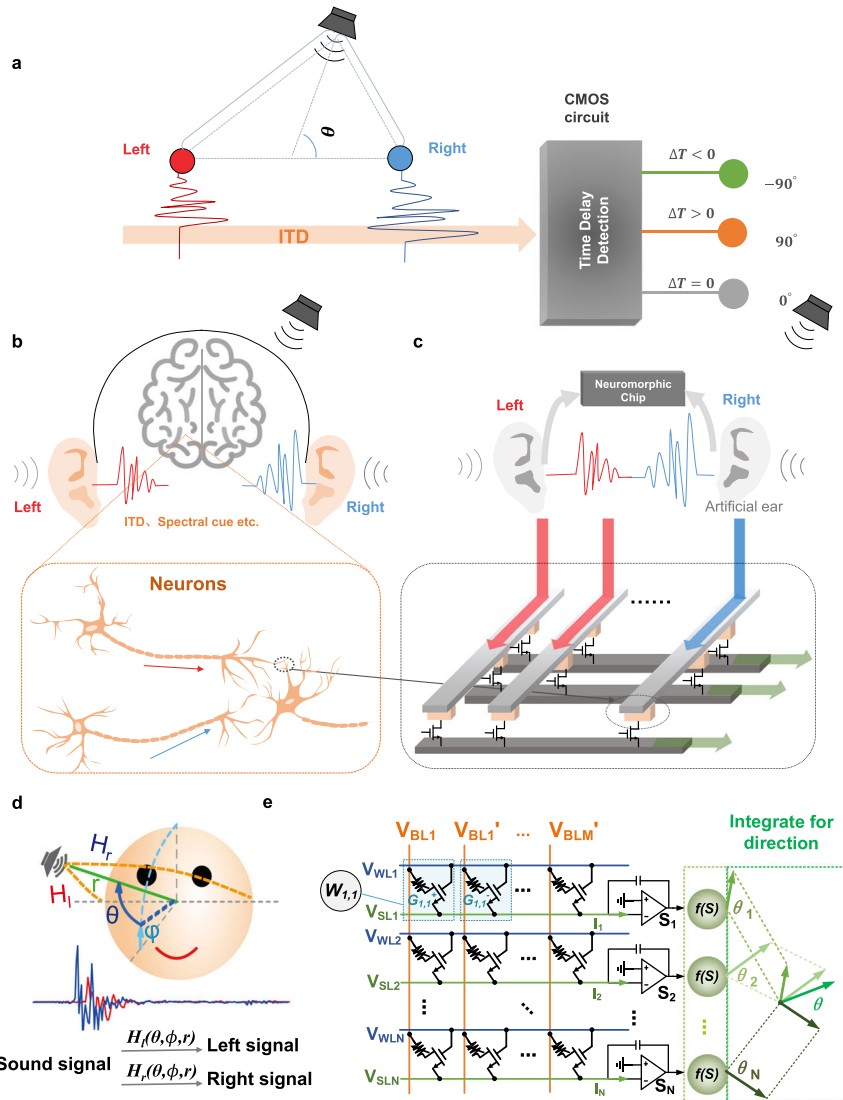

**Fig. 1 The sound localization and hardware implementation. a** The common implementation scheme for sound localization application with CMOS circuits: ITD theory. **b** Mechanism of sound localization in biological brain. Multiple binaural features are applied for neural processing to detect sound sources, including binaural time difference, spectral shape and so on. **c** A conceptual diagram of a memristor-based neuromorphic sound localization system. The memristor array acts as synapses to deal with binaural sound signals. **d** Sound source localization, HRTF and signal sample in the time domain. The transformation functions $H_l$ and $H_r$ are determined by the sound source azimuth angle $\theta$, elevation $\phi$, and distance $r$. **e** A schematic diagram for neural network implementation with an integrated memristor array. Since the weight in the neural network could either be a positive or negative value, it is mapped to two memristor cells as a differential pair.

are different from analog information processing. Typically, in a classification application, each output neuron corresponds to a certain label and refers to a discrete value. The main target is to maximize the "right" one. The neural networks used for these tasks are redundant[35]. As a result, in many complex cases, a lot of output nodes are desired, posing a great challenge to hardware overhead and energy consumption. In the sound localization task, a representative analog computing application, both the input signals and the output detected direction are continuous values. When a neural network is employed for sound localization, the output neurons work together to obtain for the predicted direction. Benefits from the significant reduction of the network scale, the hardware cost is much smaller than a classification network. However, since every synapse and neuron influence the output result, this analog computing task tends to be susceptible to errors. In addition, for training samples corresponding to various directions, the supervised values of the target neurons are different. Therefore, in the feed-forward and training process, this task is less tolerant of inevitable non-ideality of hardware, putting forward higher requirements for device performance and weight update approach. Although several weight update schemes have been proposed and successfully demonstrated in memristor arrays[26,36,37], further efforts to improve the training strategy are still required for analog computing tasks. To date, the hardware implementation of sound localization with high efficiency and in situ training is still challenging.

In this work, with analog computing-in-memory characteristics, we develop a memristor-based brain-like algorithm and architecture, capable of handling complete sound signals received by two artificial ears, as shown in Fig. 1b, c. With the integrated 1 K HfO$_x$-based analog memristor array, a subset of Head Related Transformation Function (HRTF) dataset is in situ trained based on the neural network architecture. The brain-like learning algorithm of the sound localization function is successfully realized on the memristor array with a proposed in situ training method, namely, a multi-threshold-update scheme. The tradeoff between training results and hardware overhead with different training schemes and different hardware platforms is further analyzed.

## Results

**Sound localization algorithm.** Sound localization is a basic cognitive function of human beings and a typical example dealing with analog information. As presented in Fig. 1d, the relationship between the sound source and binaural soundwaves, known as the Head Related Transform Function (HRTF)[38,39], describes distance $r$, azimuth angle $\theta$ and elevation $\phi$ contributing jointly to the differences of the sound signals received by two ears. Previous works only dealt with differences in binaural features, such as time or intensity differences. Instead of this partial information, we train a neural network to derive the relationship between angle $\theta$ and two complete serials of sound signals, suitable for implementation in the memristor array.

The received binaural soundwaves are transformed to the Fourier domain, and the supervised outputs are designed as Gaussian curves to describe the probability of sound source localization (Supplementary Fig. 1a, b). The neuron outputs work together to determine the detection of binaural sound signals. As presented in Eq. (1), the supervised outputs $y_{\text{teacher}}(\alpha_i)$ are constructed by two terms, a probability in Gaussian form and a correction term for better detection, in which $\alpha_i$ represents the output channel and ang denotes the actual angle of the sound source. To fit in the network, $\alpha_i$ is sampled from $-120°$ to $120°$ with a step of 40°. Some generated teacher outputs are illustrated

in Supplementary Fig. 1c.

$$y_{\text{teacher}}(\alpha_i) = \exp\left(-\frac{(\alpha_i - \text{ang})^2}{2\sigma^2}\right) \cdot \left(1 + \left(\frac{\alpha_i}{120°}\right)^2\right) \quad (1)$$

To determine the final predicted angle $\alpha$, the network outputs are treated as vectors, where the output values contribute to the module while the represented direction contributes to the phase. The angle vectors $\alpha_i$ are summarized together weighted by the corresponding network outputs (Supplementary Fig. 1d). In this way, the larger the network outputs are, the more likely the sound source direction is and the larger the contribution. Different from classification tasks, all neurons affect the final result, even small outputs, as expressed by Eq. (2).

$$\alpha = \sum_{i=1}^{7} y_{\text{output}}(\alpha_i) \cdot \alpha_i \quad (2)$$

As shown in Fig. 1e, for the hardware implementation on a 1T1R memristor array, considering that the calculated weight could either be a positive or negative value, it is divided into positive and negative values and mapped to two adjacent cells on the memristor array. In the training process, the synaptic weights are trained with the gradient descent method. In update cycles, the positive or negative weight cell is randomly selected for weight update. The description of the CIPIC HRTF dataset[38] can be found in the CIPIC dataset section. More details are shown in the sound localization algorithm in the methods section.

**Memristor device.** Since the conductance of each memristor contributes to the predicted direction, every single device plays a more critical role than in the classification tasks that have already been widely realized. In this case, the performance requirements for devices are higher. To demonstrate the sound localization function, an integrated 1 K analog memristor array with 128 rows and 8 columns is fabricated. As illustrated in Fig. 2a, a TiN/TaO$_y$/HfO$_x$/TiN stack structure is designed to achieve bidirectional analog behaviors. Detailed information about the device fabrication process[40] is shown in the memristor array fabrication section.

The memristor array exhibits good analog switching characteristics. With multiple pulses, the memristor cells could be programmed to conductance states between 4 and 40 μS, as shown in Fig. 2b. Applying identical pulses on the bit-lines, the conductance of the memristors in the array can be modulated in an analog way. As shown in Fig. 2c, with 1.5 V/−1.4 V 50 ns pulses, the conductance increases/decreases continuously from the lowest resistance state (LRS) to the highest resistance state (HRS) or from the HRS to the LRS. Compared with a binary memristor, there is no abrupt conductance jump in the SET/RESET process.

Due to the inherent physical mechanism, the intrinsic randomness of memristors is similar to that of biological synapses. When voltage pulses are applied on the memristor, oxygen ions move towards a specific direction and reform the Vo conductive filaments with randomness[41,42]. Figure 2d statistically quantifies this random conductance fluctuation. For any initial state, there is a broad distribution of conductance change on the memristor cells under one SET/RESET pulse. The verification scheme is the most common method to compensate for this inherent variation. However, when training a neural network with massive parallelism, it is inconvenient to program cells one by one with high precision. Inaccurate programming will cause deviation between device conductance and the target value. Brain-like randomness may help increase the diversity between memristor cells, but it will also lead to a decrease in learning

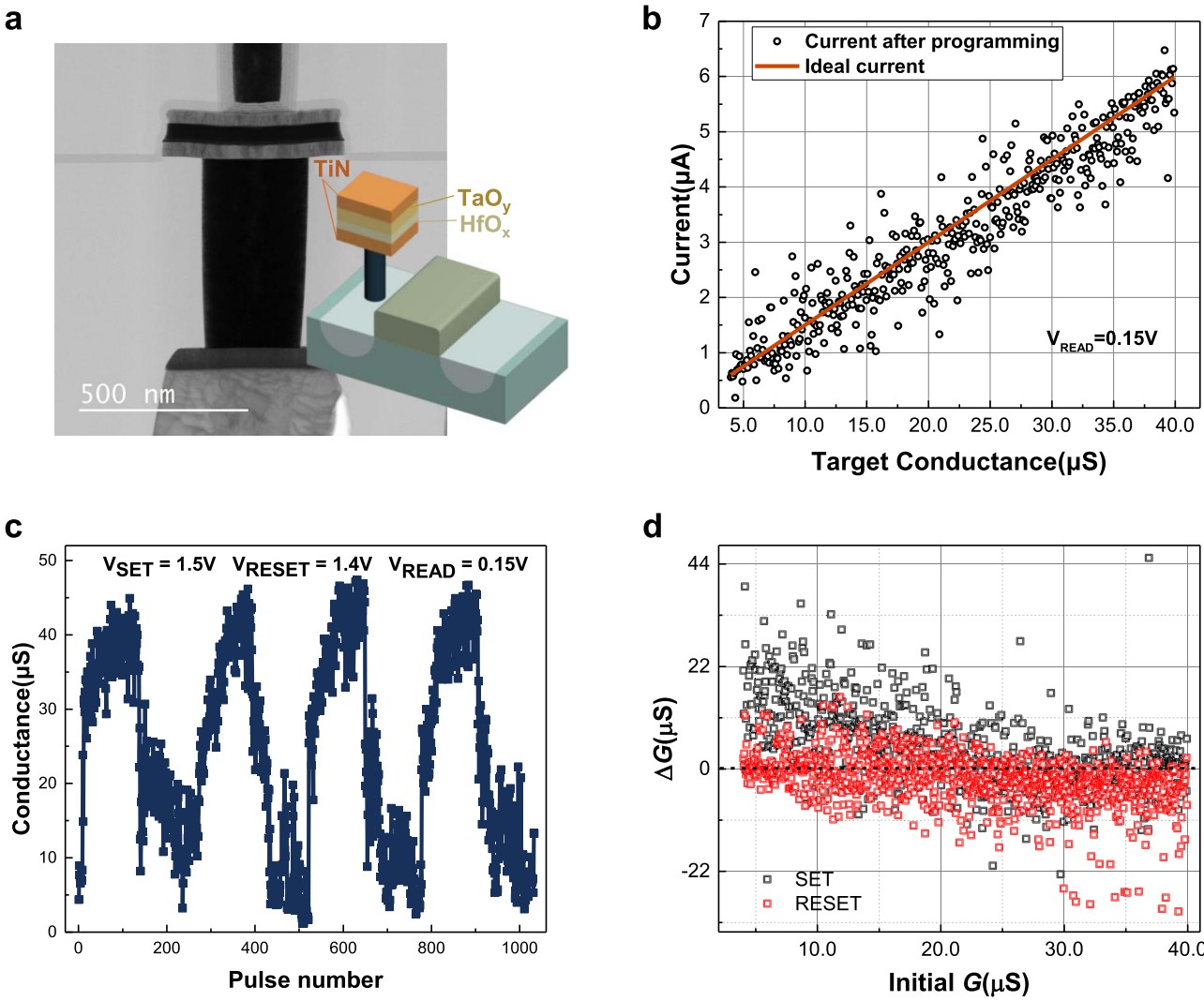

**Fig. 2 Memristor characteristics. a** The stack structure and TEM image of the memristor device. **b** With the verification scheme, the mapped conductance distributions of 420 memristors with the target conductance states between 4 and 40 μS. **c** Bidirectional analog switching behavior: device conductance increases/decreases continuously with the number of SET/RESET pulses. SET pulse: 1.5 V, 50 ns; RESET pulse: 1.4 V, 50 ns. In every cycle, the standard deviation of $\Delta G$ under the same SET/RESET pulse is 2.64 μS. **d** The conductance changes after applying one SET/RESET pulse. For SET and RESET process, the average $\Delta G$ values are 4.12 and −2.44 μS.

efficiency and accuracy, raising challenges for the training method.

**In situ training schemes for sound localization.** Through learning, the human auditory system has the ability to adapt to changes[1]. In the ex-situ training scheme that has been widely employed in previous works, the weight calculated by software remains unchanged[25]. The ex-situ training is easily affected by non-ideal parameters of hardware, for example, programming variations, state-stuck devices, conductance drift, and so on[43,44]. By adjusting weight values according to the real-time output deviation, the in situ training method[44] could adapt well to environmental changes. In the previous studies, two schemes have been proposed to update the conductance value during the in situ training process[26], namely, in situ with verification and without verification (Supplementary Fig. 2). In the first case, the network is trained accurately but not in an efficient way. It has been shown that approximately 40 pulses are required to achieve 4-bit precision, which largely limits the operation speed[45]. In addition, in an example of the write strategy, additional circuits

are designed, including a complex comparator[46]. Thus, there is a tendency that the verification needs much more chip area and time cost in hardware implementation[46–48]. For the second case, only a programming pulse is given on the basis of the sign of the calculated update values. The overhead is greatly reduced at the cost of a certain training accuracy loss. However, this simplified scheme is not applicable for sound localization. To illustrate, with this scheme, the distribution of weight update values of various angle samples is simulated, as shown in Fig. 3a. For some large update values, the conductance change after one SET/RESET pulse is far from reaching the target value. On the other hand, for a small update value, one pulse may change the conductance much larger than the targeted value. It is difficult to find a universal pulse operation condition to update all weights as similar as possible to the desired values, demanding weight adaptation in subsequent cycles.

To tackle the problems of the two traditional methods mentioned above, a multi-threshold-update scheme is developed. As depicted in Fig. 3b, from bottom to top, there are several thresholds to determine the number of pulses applied on the memristor device. These thresholds divide the weight update

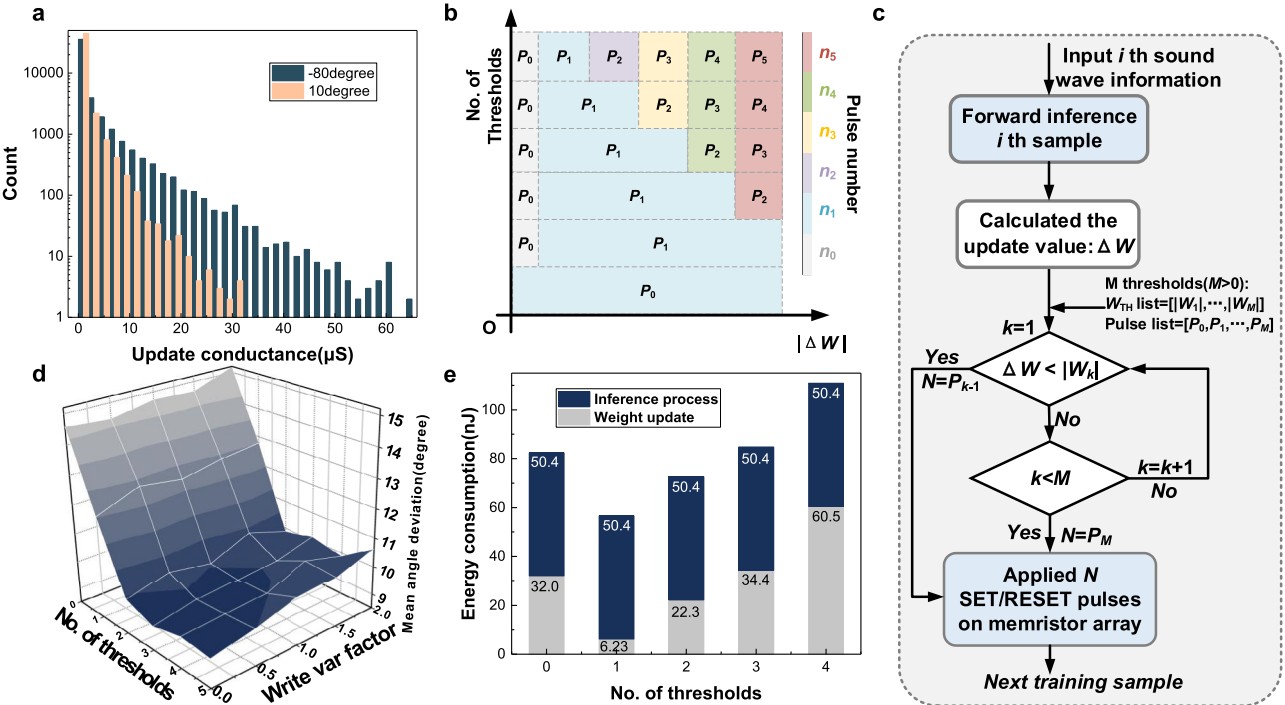

**Fig. 3 Schematic diagram and simulation results of various training schemes. a** The distributions of update values of samples corresponding to various angles. **b** Schematic diagram of the multi-threshold-update scheme. As the updating levels increase, there will be more thresholds for $\Delta W$ classification to apply different numbers of pulses on the memristor device. **c** Flow chart of in situ training with the multi-threshold-update scheme. **d** Simulation results under various update levels and variation factors. The factor scales the updating variation of the devices. **e** The total energy consumption of memristor-based sound localization under various number of update thresholds.

value ($\Delta W$) into intervals filled with various colors. In the color bar, the numbers annotate pulse numbers used in that case. With this comparison rule, Fig. 3c shows the flowchart of the update stage. After the feed-forward and weight calculation, the calculated $\Delta W$ is compared with thresholds and distributed into different update levels. The number of update pulses is determined based on the comparison results. The training schemes section presents a detailed description.

To prove the feasibility and optimize the training performance, we simulate the in situ training results with the multi-threshold-update scheme for sound localization. The memristor model is depicted in Supplementary Figs. 3 and 4. Figure 3d presents the average angle deviation under various training methods. With the zero-threshold-update scheme, also known as the conventional scheme without verification, the trained network shows a large deviation. The introduction of more update levels improves the average training accuracy by approximately 4–5 degrees compared to the without verification scheme. However, as the number of thresholds is further increased, the training accuracy will not be improved to a greater extent. Conversely, the unfavorable effect of writing variation becomes more obvious.

On the other hand, more thresholds would increase the hardware overhead. The hardware cost of various in situ training schemes are studied. The energy consumption of inference and training processes are calculated as the product of update times and energy consumed each time. Figure 3e, Supplementary Fig. 5a, and the benchmark for hardware implementation section present the simulated total energy consumption and processing time cost in the memristor array under various number of thresholds. As illustrated in Fig. 3d), for the two-threshold-update scheme, the total number of update pulses during the training process is reduced compared to the without verification scheme, as well as the operation cost. The training result is also better with

this scheme. As a result, when considering the hardware complexity, as well as the simulation results, in situ training with the two-threshold-update scheme is the most suitable method for the sound localization task in this work. The parameters in the update scheme are determined based on the device characteristics and algorithm. It goes as follows: one pulse is applied if the conductance update value $|\Delta G|$ falls in $|10\,\mu S| > |\Delta G| > 1\,\mu S|$, while successive 150 pulses are applied for $|\Delta G| > 10\,\mu S|$. No pulse is applied if $|\Delta G| < |1\,\mu S|$. Different types of devices also follow the similar trend, as presented Supplementary note 7. This method also applicable for more complex tasks, as shown in Supplementary Fig. 5b. Depending on the device characteristics and tasks, the thresholds and pulse number may be different in other works.

## Results and discussion

In the integrated 1 K memristor array, the in situ learning of sound localization function is experimentally demonstrated. Supplementary Fig. 6 and the measurement platform section present detailed information about the test platform. Overall, the results indicate the feasibility of this analog computation implementation on the memristor array. Figure 4a, b show the weight distributions and statistical results under various schemes. For in situ training with the introduction of thresholds, most weights are small values, similar to the ideal weight matrix, while the weight distribution under the conventional without verification scheme is quite different. In that case, most of the negative weights are located in the rows corresponding to the large angles. For large-angle samples, the neuron outputs that should have large contribution are small, leading to unsatisfactory prediction results, as shown in Fig. 4c, d. With regard to the multi-threshold-update scheme, the comparison operation ignores small weight update values of approximately 0. Compared to the former

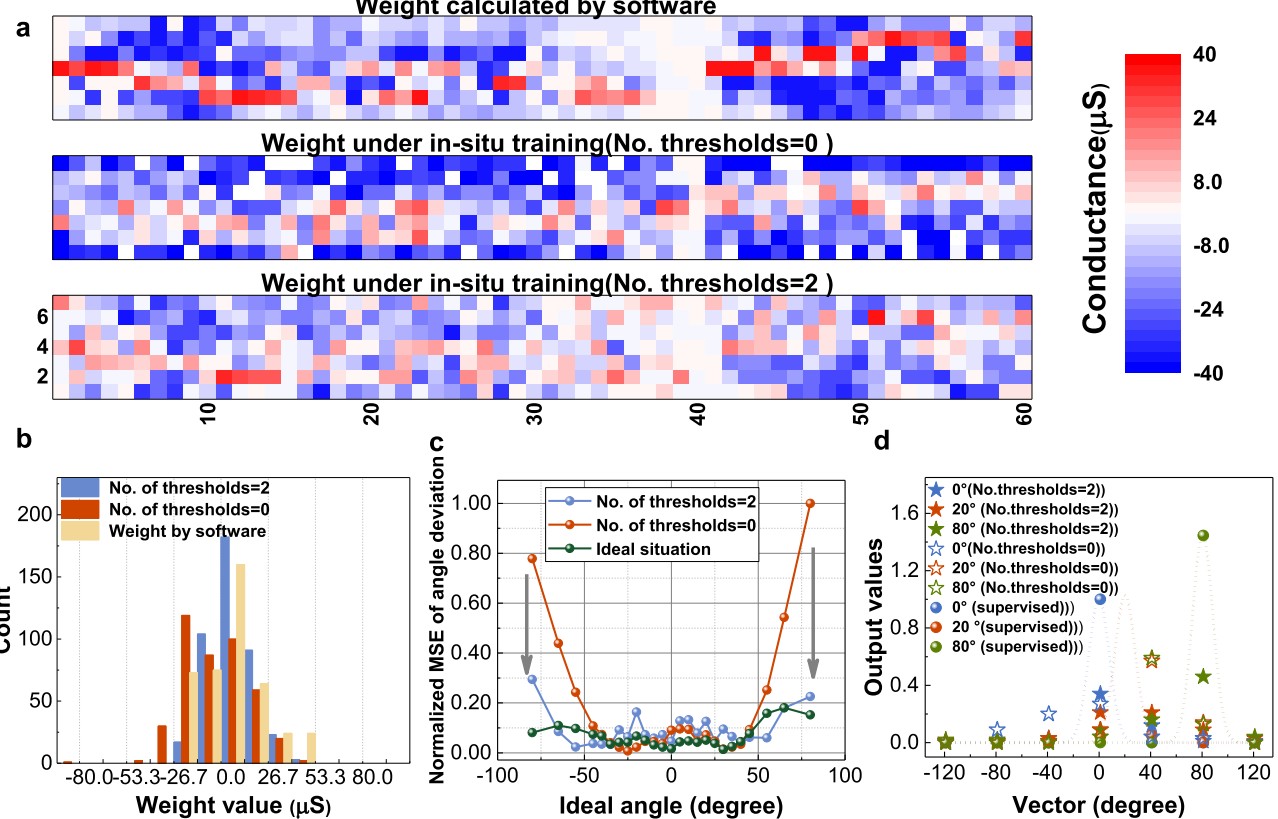

**Fig. 4 Experimental demonstration results of sound localization. a** The ideal weights and the measured trained weights under the number of thresholds = 0 and 2. **b** The weight statistical distribution results under various in situ training schemes and ideal weights. **c** The normalized mean square error (MSE) of the experimental results under the number of thresholds = 0 and 2. **d** The ideal and experimental outputs of three angle samples with various training schemes.

method, this scheme makes the trained weight distribution more stable and more inclined to gradually approach the optimal solution. It is suitable for this analog computing task and decreases the MSE of the experimental results by ~45.7% for in situ training, as shown in the Supplementary Table 1.

Furthermore, we evaluate the hardware performance of a larger memristor-based sound localization task and compare it with previous works. A schematic diagram of the proposed memristor-based implementation circuit is presented in Supplementary Fig. 9. The network has two layers with the sizes of $200 \times 50$ and $50 \times 11$, respectively. The most peripheral circuits and memristor array are simulated by the simulator XPEsim under a 130 nm technology process[49]. The performance results of the integrator and analog-to-digital converter (ADC) are obtained through testing and reference[37,50]. The evaluation details can be found in the benchmarking for hardware implementation section and Supplementary Tables 2 and 3. Benefitting from computing-in-memory, with small accuracy degradation (<1.5 degrees), the memristor-based sound localization reduces the energy consumption by ~184× compared to the existing application specific integrated circuit (ASIC) designs with conventional architecture[6]. In the future work, by optimizing the weight-update nonlinearity, variations and other non-ideal effects of memristors, the training accuracy can be further improved.

As a representative neuromorphic computing application, both the input signals and the detected results of the sound localization task are analog values. For the hardware implementation of such a cognitive task, a promising update scheme with high efficiency and accuracy is developed and experimentally demonstrated in the 1 K oxide analog memristor array in this work. This scheme

yields improvements of 45.7% in the training performance compared to the previous method. The hardware performance is also analyzed, where memristor-based sound localization reduces energy consumption by ~184× relative to existing ASIC design. This work paves the way towards a cognitive system that rivals the CMOS design in processing accuracy and hardware costs.

## Methods
**Sound localization algorithm**. The hardware implementation of the sound localization network is described in Fig. 1e. In this demonstration, the CIPIC HRTF dataset is used for training and testing the neural network[38]. The network contains 60 input and 7 output neurons. The network model and mapping scheme used in this work are expressed as Eq. (3):

$$y_j = \sigma\left( \sum_i^m w_{i,j} \cdot x_i + b \right) = \sigma(s_j + b) \tag{3}$$

To implement matrix-vector multiplication in the memristor array, in this experiment, the input values are quantized to 16 levels and presented as multiple read pulses, as expressed in Eq. (4).

$$v_{\text{out},j} = \int_0^{n_i} \sum_i v_{\text{read}}(t) \cdot (G_{i,j}^+ - G_{i,j}^-) dt \tag{4}$$

$w_{i,j}$ represents the weight of network. $x_i$ is the $i$th input value and represents the integral time $n_i$. Considering that the calculated weight could either be a positive or a negative value, it needs to be divided into positive and negative weights and mapped to two memristors. In the feed-forward process, the multi-pulse read process is also divided into two processing steps. In each stage, the transformed binaural signals are fed into the BLs corresponding to the positive and negative weight cells. The neuron outputs, $v_{\text{out},j}$ are obtained by subtracting the current flowing through the SLs in two stages. The $s_j$ in Eq. (3) is substituted with array output $v_{\text{out},j}$. Analog neuron outputs contribute for the prediction angle together.

For the single-layer sound localization, the gradient descent method is employed to calculate update weight matrix, as represented as Eq. (5).

$$dW_{i,j} = \eta \cdot x_i \cdot (y_j - \widehat{y}_j) \cdot \frac{\partial y_j(x_i)}{\partial x_i} \qquad (5)$$

In the training process, $\eta$ and $x$ are respectively the learning rate and inputs. $y$ and $\widehat{y}$ refer to the actual and target outputs. In the experiments, minibatch is chosen as 5. Therefore, in each update iteration, the update weight is calculated as the average value of 5 feedforward calculation results.

**CIPIC dataset**. The dataset used in this demonstration is a subset of the CIPIC HRTF dataset[38]. It provides a collection of binaural signals from the same sound source for HRTF research. Samples are randomly selected from the dataset. We choose samples with azimuths ranging from −90° to 90° and elevations less than 15° for the experiments.

**Memristor array fabrication**. To achieve bidirectional analog conductance modulation behavior, a TiN/TaO$_y$/HfO$_x$/TiN stack structure is adopted with the material proportion delicately designed. Transistors are fabricated in the standard CMOS foundry. The length and width are 1 and 0.5 μm, respectively. The contact and the first TiN layer are also prefabricated by the foundry. The TiN, TaO$_y$, HfO$_x$, and TiN layers are sequentially deposited with atomic layer deposition (ALD). The 8 nm-thick HfO$_x$ acts as the switching layer. The 45 nm-thick TaO$_y$ layer acts as thermal enhanced layer, whose low thermal conductivity confines the heat in the HfO$_x$ layer, inducing the increasing temperature of conductive filament (CF) region and more uniform distribution of oxygen vacancies[51]. Therefore, this memristor stack provides resistive switching with good analog characteristics. Transistors could provide better control of the analog behaviors of memristors. The 1024 one-transistor-one-resistor (1T1R) cells are grouped into 128 rows and 8 columns. The 128 word-lines are connected to the gates of transistors. Memristor devices are located between bit-lines and drains of transistors.

**Measurement platform**. The measurement platform is shown in Supplementary Fig. 6, and the 1 K memristor array is connected to the probe card. During the training process, the tester generates SET/RESET/READ pulses and sends them to a probe card that is connected to the array. The current flow through SLs is collected by the probe card and sent back to the tester. A PC sequentially performs further analysis. The activation function and other processing operations are executed in the software. In the SET (conductance increase) process, 1.5–2 V is applied on the transistor gate to limit the current. A voltage pulse (1.5 V, 50 ns) is applied on the BL, while the SL is connected to 0 V. During the RESET (conductance decrease) process, a voltage pulse (1.6 V, 50 ns) is applied on the SL. Moreover, 0 V and a high voltage to ensure that the transistor opened are applied on the BL and WL, respectively.

**Training schemes**. The schematic of the multi-threshold-update scheme can be seen in Fig. 3b. From bottom to top, there are operation schemes corresponding to various thresholds. The color in different ranges of $|\Delta W|$ corresponds to the programming pulse number used in that case ($n_0 < n_1 < n_2 < n_3 < n_4 < n_5$, $n_0$ and $n_1$ are generally 0, 1). For example, when the number of thresholds is 0, the pulse list is $[n_1]$. If $\Delta W > 0$, one SET pulse is applied. Conversely, if $\Delta W < 0$, a RESET pulse is required. When the one-threshold-update method is applied, the pulse list is $[n_0, n_1]$. Only if $|\Delta W|$ is greater than $|W_1|$, one SET or RESET pulse is applied to the corresponding cell.

Figure 3c represents the flow diagram of the multi-threshold-update scheme. The weight update value is calculated after the feed-forward and error calculation. For $M$ thresholds, $W_{TH} = [|W_1|, …, |W_M|]$, and the pulse number list $= [P_0, …, P_M]$. These thresholds divide the update value into multiple intervals. The calculated $|\Delta W|$ is sequentially compared with $|W_1|, …, |W_M|$. According to the intervals at which the update weights are located, the number of update pulses is determined based on the comparison results. For example, when $M = 2$, the $|\Delta W|$ values in $[0, |W_1|)$, $[|W_1|, |W_2|)$, $[|W_2|, +∞)$ correspond to the $P_0, P_1, P_2$ update pulses to be applied to the memristor device ($P_0 = n_0$, $P_1 = n_1$, $P_2 = n_5$). The positive and negative memristors are randomly selected as the updated device. The updated conductance value is closer to the target value. The parameter analysis is presented in Supplementary note 7.

**Benchmarking for hardware implementation**. We evaluate the hardware overhead of sound localization. In the evaluation process, the main circuit modules are all taken into consideration, consisting of the memristor array, integrator, ADC, drivers, and shift adder. The synaptic weight matrix is mapped to a 2T2R memristor array. In the feed-forward process, the transformed signals are converted to two opposite voltage pulses and applied on the BL$^+$ and BL$^−$. Performance of the BL driver, WL drivers and integrators comes from test results of actual circuits[37]. The shift adder module is simulated by XPEsim under 130 nm technology node[49]. Data of the actual Fourier transformation circuit are obtained through reference[52]. The detailed results can be observed in Supplementary Figs. 9 and 10 and Supplementary Table 3.

## Data availability
The data other than that provided in Source data that support the findings of this study are available from the corresponding author upon request. Source data are provided with this paper.

## Code availability
The codes that support the findings of this study are available from the corresponding author upon reasonable request.

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

## Acknowledgements

This work was supported in part by the MOST of China 2021ZD0201200 (H.W.), NSFC 62025111 (H.W.), 61874169 (B.G.), 92064015 (X.L.), XPLORER PRIZE (H.W.), the IoT Intelligent Microsystem Center of Tsinghua University-China Mobile Joint Research Institute, and the Beijing Advanced Innovation Center for Integrated Circuits. We acknowledge the use of the CIPIC Database.

## Author contributions

B.G., Y.Z., Q.Z., S.Z., and H.W. proposed and designed the experiment. Y.X. and X.L. fabricated the memristor array. Y.Z. and S.Z. performed the measurement. Q.Z., W.Z., and P.Y. contributed to the algorithm and simulation. Y.Z. and Q.L. made the benchmark. B.G. and Y.Z. analyzed data and wrote the manuscript. B.G., H.Q., and H.W. supervised the project. All authors discussed and reviewed the manuscript.

## Competing interests

The authors declare no competing interests.
