## [Peer Review File · Nature Communications]

Reviewers' Comments:

Reviewer #1:

Remarks to the Author:

The authors have presented an interesting use case for memristive based computing. Authors have shown fabrication, characterization and simulation results. However it is important to address the below concerns to enhance the manuscript.

Lines:225-227:

1.Please specify the observed MSE in result table format for the two methods even if it is obtained from simulation. Also, it will be advantageous to compare the result to an ideal quantized network without considering memristive devices to understand the accuracy trade-off.

Lines 269:

2.The experiment shown is for a single layer. Through simulations or experiments can the authors comment on the impact of using multi-layer networks. It will be valuable to analyze the case of multi-layer networks as well.

Section on Training:

3.How do asymmetric weight updates affect the training in terms of simulated accuracy? It would further strengthen the analysis if the authors can provide a parameter exploration for number of thresholds, non-linearity of devices, etc. parameters and show its impact on- accuracy, energy, latency of the network .

Lines 330-332:

4.Please explain in more detail which exact circuits are borrowed from which reference. Can the authors comment how the difference in operating condition ranges used for this study will impact the circuit performance compared to the original references.

5.Figure 2: please specify the measurement statistics/count for 2(b),2(c) and 2(d).

6.Figure 3(e): please provide the details of energy calculation methodology followed.

Benchmarking

7.One concern is that the authors have benchmarked w.r.t an ASIC reported in 2009 which was fabricated at 500nm process node. It will be better if the benchmarking can be updated w.r.t more recent low-power hardware for the application in question.

8.

There are several works in literature on neuromorphic CMOS silicon cochlea circuits, some even showing sound source localization. It will be helpful if the authors can also include and compare against these works their proposed technique based on memristive devices. And show where memristive circuits are bringing the true benefit.

9. Please provide details of the selector device and RRAM set/reset current values.

10. Can the authors comment on the impact of device endurance on the proposed methodology?

11. Can the authors comment on the impact of RRAM intermediate resistance state retention for the proposed methodology?

12. Which type of resistive switching mechanism is at play in the devices the authors have used? This information should be discussed in the paper.

13. Lines 189-191: Can the authors explain how this helps to increase the accuracy?

Supplementary Sheet

14. Supp Fig 3: Can the authors provide equations and reference for the device model used?

15. Supp fig 4: Can the authors provide comparison of the simulated curves and the measured curves of the real fabricated devices with both D2D + C2C variation.?

16. Supp Fig 6: Please elaborate on what is meant by tester in the middle of the picture.

17. Supp Fig 8 and Tab 1:

Please specify the precision of the output localization range as that is essential. The presented solution has limited precision of upto 40 degrees compared to the CMOS solution which has a resolution of 2 degrees. Can the authors comment on that aspect?

18. Supp Fig. 7: it is not clear which elements were fabricated in CMOS which all were simulated. Can the authors please clarify this.

19. For the RRAM device can the authors share more details like layer thickness etc.

Minor Comments

1. Please re-verify the formatting of all references. Title is missing in some places.

2. fig 1c is indicating selector free crossbar?

3. In figure 4 some axis labels are too small to read.

Reviewer #2:

Remarks to the Author:

The paper by Zhou et al. presents a memristor-based implementation of sound localization with ANN. The paper is well written but I believe this work represent only an incremental improvement in this field.

- It is not clear what is the key advantage of using memristor based VMM for this particular sound localization task. The authors claim that the key challenge of this application is to have a "vote" at the output layer, but this seems to me to be what you are doing if you consider the same classification task with a 2 layer (or more) neural net. Also, the challenge of having multiple output neurons to optimized (and not only one) is only due to the fact that the authors are using a small 1 layer network. This is what is expected when considering the difference between classification and pattern matching.

- The second major claim is to have a 47% improvement of performances with 177% reduction in energy consumption. OK, but could you compare the improvement in energy consumption for the same performances? It seems to me that 46% is for the particular hardware of the study (i.e. different threshold methods) and 177% is in between different hardware.

- The most interesting part of the paper is the threshold method used to circumvent the non linear weight update and low accuracy of memristor tuning. This aspect would deserve a more complete analysis such as accuracy improvement with respect to other technics, linearity of the weight updates, mapping between threshold and number of pulses for update, etc...

- why low accuracy memrsitor can implement in-situ training while it is known that resolution during in-situ training is a strong limitation for application: is it only task related or is it thanks to the threshold method?

Reviewer #3:

Remarks to the Author:

Summary: In this paper, a scheme is presented for training a memristor-based sound localization network in situ that uses thresholds of ΔG to determine how many SET or RESET pulses should be applied to a memristor weight. This helps compensate for the stochastic change in conductance for a given initial conductance, which reduces the likelihood that the change in conductance will be too large or too small. This scheme is compared to previous schemes, where the change in conductance is either verified (requires more energy to double-check) or left unverified (less accurate).

Strengths: The paper is technically sound, describing the memristor-based network operation in detail and supporting results with plotted data. The driver, ADC, inverter, etc. circuits are considered when estimating the power. A 45.6% improvement in accuracy and a 177x decrease in energy consumption are reported, which are credible and impressive results.

Impact: As previously mentioned, I believe that in situ training is an important topic in analog computing because of the nonvolatile nature and interesting dynamics of memristors. Sound localization is just one of many potentially useful applications. However, this work seems to focus on updating memristor values accurately and efficiently. This goal is important, though, with such emphasis on in situ training, some novelty in the training method (how is the ΔG calculated?) might further improve the impact of this work.

Thoughts: This work produced promising results (45.6% improvement in accuracy and 177x decrease in energy consumption) in the important field of in situ training. I also feel that while the work is technically sound and novel, it does not seem particularly technically difficult. I think that if the focus is on in situ training specifically instead of just efficient/accurate memristor programming, work should also perhaps be done in terms of calculating gradient or implementing backpropagation in general. In conclusion, I would accept this paper, but with slight modifications to the motivation section.

Reply to Reviewers' Comments:

We would like to thank the reviewers for the comments. According to the comments, we have made the revisions on the manuscript. Here is a point-by-point reply to the Reviewers' Comments.

Reviewer #1:

The authors have presented an interesting use case for memristive based computing. Authors have shown fabrication, characterization and simulation results. However, it is important to address the below concerns to enhance the manuscript.

1. Lines:225-227: Please specify the observed MSE in result table format for the two methods even if it is obtained from simulation. Also, it will be advantageous to compare the result to an ideal quantized network without considering memristive devices to understand the accuracy trade-off.

Reply: We sincerely thank the reviewer's effort and valuable suggestions to help us improve our manuscript. **Table R1** reveals the normalized mean square error (MSE) in result table format for the experimental and ideal quantized results. The proposed multi-thresholds-update scheme yields improvements of 45.7% in the training performance compared to the previous without-verification scheme (No. thresholds=0). Besides, the ideal quantized network results have been added in the **Fig R1**, including W_{bit} (weight precision) =1&2, O_{bit} (output precision) =3. The *in-situ* training scheme could alleviate the accuracy degradation caused by the low precision.

Situation	No. of thresholds=2	No. of thresholds=0	Ideal quantized network ($W_{\text{bit}}=1$, $O_{\text{bit}}=3$)	Ideal quantized network ($W_{\text{bit}}=2$, $O_{\text{bit}}=3$)
Average normalized MSE of angle deviation	0.0927	0.171	0.118	0.109

Table R1 The average normalized mean square error (MSE) of the experimental results under the No. of thresholds = 0 and 2 and ideal quantized networks.

Fig R1 The normalized mean square error (MSE) of the experimental results under the No. of thresholds = 0 and 2 and ideal quantized networks.

Table R1 has been added in the revised version. Please see the **Sup Table 1**. “It is suitable for this analogue computing task and decreases the MSE of the experimental results by ~45.7% for *in-situ* training, as shown in the **Sup Table 1**” (page 9, line 2).

- Lines 269: The experiment shown is for a single layer. Through simulations or experiments can the authors comment on the impact of using multi-layer networks. It will be valuable to analyze the case of multi-layer networks as well.

Reply: We thank very much for valuable comments by the reviewer. In the simulation, a two-layer neural network for sound localization is established, as shown in **Fig R2(a)**. The numbers of input, hidden and output neurons are respectively 80,40,7. The analogue output neurons in the last layer work together to contribute for the final prediction result. To prove the feasibility of proposed method, we simulate the *in-situ* training effect of the two-layer neural network with various training schemes. Using the two-thresholds-update scheme, the training accuracy of the two-layer network is 5.1% higher than that of the single-layer network. With the introduction of update thresholds, the average angle deviation of trained network is reduced by 61.5%, as depicted in **Fig R2(b)**. Therefore, the proposed method also delivers significant performance improvement of training accuracy for the multi-layer network.

Fig R2 (a) Schematic of two-layer sound localization network. The number of input, hidden, output neurons are respectively 80, 40, 7. The output neurons contribute for the final prediction angle. (b) Training results of the two-layer sound localization network with different number of update thresholds.

We also have added simulation results of two-layer sound localization network with multi-thresholds-update scheme (page 8, line 13). “This method also applicable for more complex tasks, as shown in **Sup Fig. 6** and **Sup Fig. 7**. Depending on the device characteristics and tasks, the thresholds and pulse number may be different in other works.” Please see the revised manuscript.

- Section on Training: How do asymmetric weight updates affect the training in terms of simulated accuracy? It would further strengthen the analysis if the authors can provide a parameter exploration for number of thresholds, nonlinearity of devices, etc. parameters and show its impact on- accuracy, energy, latency of the network.

Reply: Thank you very much for your detailed comments. We discuss the *in-situ* training performance, energy consumption, processing time with different device characteristics and update schemes. The simulation situations are the same as those used in the experiment. The initial conductance state is 4 μ S.

Detailed results are shown below:

1) Asymmetry

Fig R3(a) illustrates memristor models with asymmetric and symmetric update behaviors. Using the multi-threshold-update scheme, training accuracy is presented in **Fig R3(b)**. It can be found that different types of devices follow the similar trend: the

accuracy increases as the number of thresholds increase, but tend to saturate as when the number is greater than 2.

With the same zero-threshold-update scheme, the training effect with the asymmetric model is a little better than the symmetric device. For symmetric devices with low conductance values (see black line in Fig. R3(a)), a SET/RESET pulse will lead to a large conductance change (large learning rate). During the *in-situ* training process, conductance of many key devices fluctuates in the low conductance range and hardly reaches to the target value (Fig R4(a)). It differs from the training process with asymmetric device (Fig R4(b)) and causes a loss of training accuracy. In contrast, although the asymmetric devices with low conductance values have a large conductance change with a SET pulse, they can have a small conductance change with a RESET pulse, resulting in the improvement of accuracy. In this task, compared to initial state $4\mu\text{S}$, random initialization is more conducive for the symmetric device model.

Fig R3(c-d) present the hardware overhead of the sound localization network with various *in-situ* training schemes, including processing time and energy consumption. In accordance with the conclusion in the manuscript, considering the hardware cost and training results, two-threshold-update-scheme is more appropriate for symmetric and asymmetric devices.

Fig R3 The impact of asymmetric behavior on the performance of memristor-based sound localization. (a) The device model with asymmetric and symmetric behaviors. (b) Training accuracy with various number of thresholds. (c) Processing time spent on the memristor array during *in-situ* training process. (d) Energy consumption spent on the memristor array during

in-situ training process

Fig R4 Conductance change of 6 cells with zero-threshold-update scheme during *in-situ* training process. (a) Symmetric model. (b) Asymmetric model.

2) Nonlinearity

As illustrated in **Fig R5**, we simulate the training accuracy with linear and nonlinear device models. For different nonlinearity characteristics, different training schemes are adopted. The thresholds and pulse numbers are the best solutions currently explored for various models. It is worth noting that with the introduction of one-update-threshold, the nonlinear device shows a better training result compared to the linear device. For the nonlinear analogue behavior, with the low initial state, the increase of conductance is greater than that of the linear model. As presented in **Fig R6**, it exhibits a wider range of conductance, making a higher training accuracy.

When the number of thresholds is greater than 2, the training results of sound localization network tend to be saturated. The linear model shows a slightly better training effect than the nonlinear model. **Fig R5(c-d)** present the hardware cost for nonlinear and linear switching models with multi-thresholds-update scheme. Considering hardware complexity and training results, the introduction of two thresholds is the more conducive for different models.

Fig R5 The impact of nonlinearity behavior on the performance of memristor-based sound localization. (a) The device models with different behaviors. Curves 1,2,3 respectively reflect device models with increasing nonlinearity values. Model 2 is closer to the actual device. (b) Simulated training accuracy with multi-thresholds-update scheme. (c) Processing time spent on the memristor array under various number of update thresholds. (d) Energy consumption spent on the memristor array under various number of update thresholds and device models.

Fig R6 Conductance change of 100 cells in first weight array. (a) (b) (c) respectively correspond to nonlinearity value 1,2,3 with the one-threshold-update scheme.

We have added more analysis in the revised **Supplementary note 8**. Please see the revised supporting information and **Sup Fig. 8** and **Sup Fig. 9**. “The impact of asymmetric behavior on the performance of memristor-based sound localization. (a) The device model with asymmetric and symmetric behaviors. (b) Training accuracy with various number of thresholds. (c) Processing time spent on the memristor array during *in-situ* training process. (d) Energy consumption spent on the memristor array

during *in-situ* training process”; “The impact of nonlinearity behavior on the performance of memristor-based sound localization. (a) The device models with different behaviors. Curves 1,2,3 respectively reflect device models with increasing nonlinearity values. Model 2 is closer to the actual device. (b) Simulated training accuracy with multi-thresholds-update scheme. (c) Processing time spent on the memristor array under various number of update thresholds. (d) Energy consumption spent on the memristor array under various number of update thresholds and device models.”

4. Lines 330-332: Please explain in more detail which exact circuits are borrowed from which reference. Can the authors comment how the difference in operating condition ranges used for this study will impact the circuit performance compared to the original references?

Reply: We sincerely thank the reviewer’s kind comments. In the evaluation of sound localization task, data of the BL driver, WL drivers and integrators are test results of actual circuit¹. The shift adder circuit is simulated under 130nm technology node². The actual implementation results of the CMOS-based Fourier transformation module can be observed in the reference³. The measured data of ADC comes from reference⁴. The operating conditions in the evaluation process are the same as those in the references. The scaling of parameters is not considered.

More descriptions about the hardware evaluation have been added (page 13, line 4). “Performance of the BL driver, WL drivers and integrators comes from test results of actual circuit¹. The shift adder module is simulated by XPEsim under 130nm technology node². Data of the actual Fourier transformation circuit are obtained through reference³.” Please see the revised version.

1 Liu, Q. *et al.* A Fully Integrated Analog ReRAM Based 78.4TOPS/W Compute-In-Memory Chip with Fully Parallel MAC Computing. *2020 IEEE International Solid- State Circuits Conference - (ISSCC)*, 500-502, doi:10.1109/ISSCC19947.2020.9062953 (2020).

2 Zhang, W. *et al.* Design Guidelines of RRAM based Neural-Processing-Unit: A Joint Device-Circuit-Algorithm Analysis. *2019 design automation conference (DAC)* (2019).

3 Yuan, G. *et al.* Memristor crossbar-based ultra-efficient next-generation baseband processors. *2017 IEEE 60th International Midwest Symposium on Circuits and Systems (MWSCAS)*, 1121-1124, doi:10.1109/MWSCAS.2017.8053125 (2017).

4 Paulus, C. *et al.* A 4GS/s 6b flash ADC in 0.13 μm CMOS. *2004 Symposium on VLSI Circuits. Digest of Technical Papers (IEEE Cat. No.04CH37525)*, 420-423, doi:10.1109/VLSIC.2004.1346637 (2004).

5. Fig 2: please specify the measurement statistics/count for 2(b),2(c) and 2(d).

Reply: We thank the reviewer for the detailed comments. In **Fig. 2(b)**, 420 memristor devices are programmed to conductance states between $4\mu\text{S}\sim 40\mu\text{S}$. The average deviation between the target and mapped conductance states is $2.51\mu\text{S}$. **Fig. 2(d)** shows the ΔG of these devices after a SET or RESET pulse. For SET and RESET process, the average ΔG values are $4.12\mu\text{S}$ and $-2.44\mu\text{S}$. **Fig. 2(c)** illustrates the typical analogue behavior of a memristor device. In every cycle, the standard deviations of ΔG under the same SET/RESET pulse is $2.64\mu\text{S}$.

The measurement statistics have been added in the manuscript. In the caption of **Fig. 2**, “**(b)** With the verification scheme, the mapped conductance distributions of 420 memristors with the target conductance states between $4\mu\text{S}$ and $40\mu\text{S}$. **(c)** Bidirectional analogue switching behavior: device conductance increases/decreases continuously with the number of SET/RESET pulses. SET pulse: 1.5 V , 50ns ; RESET pulse: 1.4 V , 50ns . In every cycle, the standard deviation of ΔG under the same SET/RESET pulse is $2.64\mu\text{S}$. **(d)** The conductance changes after applying one SET/RESET pulse. For SET and RESET process, the average ΔG values are $4.12\mu\text{S}$ and $-2.44\mu\text{S}$.” Please see the revised manuscript.

6. Fig 3(e): please provide the details of energy calculation methodology followed.

Reply: We thank the reviewer for the detailed comments. In the evaluation, the inference and update cost on the memristor array are taken into consideration. For a read operation and update operation of a memristor, the power consumptions are respectively calculated as the following equations (G_{avg} refers to average conductance value, V_{READ} , V_{SET} and V_{RESET} respectively mean the operation voltage):

$$(V_{\text{READ}})^2 \times G_{\text{avg}} = (0.2\text{V})^2 \times 15\mu\text{S} = 0.6\mu\text{W} \quad (1)$$

$$(V_{\text{SET}} + V_{\text{RESET}})^2 / 4 \times G_{\text{avg}} = (1.4\text{V} + 1.5\text{V})^2 / 4 \times 15\mu\text{S} = 31.54\mu\text{W} \quad (2)$$

We track the inference and update times with various *in-situ* training schemes. The energy consumption of inference and training processes are calculated as the product of update times and energy consumed each time.

We have added explanations about the energy calculation methodology in the revised manuscript (page 7, line 27). “The hardware cost with various *in-situ* training schemes are studied. The energy consumption of inference and training processes are calculated as the product of update times and energy consumed each time.” Please see the revised version.

7. Benchmarking: One concern is that the authors have benchmarked w.r.t an ASIC reported in 2009 which was fabricated at 500nm process node. It will be better if the benchmarking can be updated w.r.t more recent low-power hardware for the application in question.

Reply: We thank the reviewer for the valuable suggestion. We update the comparison table with another ASIC work. The technology node is 130 nm, same as the two-layer memristor-based sound localization system.

Design	Technology node/nm	Processing time/ms	Energy consumption/ μ J
Memristor-based	130	0.0028	0.306
ASIC ⁵	130	10	56.3
Improvement to ASIC	-	3571 \times	184 \times

Table R2 The final evaluation results for the ASIC design⁵ and memristor-based sound localization.

The hardware evaluation results have been updated. Please see the **Supplementary note 9** in the revised version. “We evaluate the hardware cost of a larger memristor-based sound localization network with 200 input, 50 hidden and 11 output neurons. Output neurons contribute for the final detection results. The synaptic weights are firstly trained by software and then are mapped and programmed on the memristor arrays with deviation of 2μ S. After that, the final layer is online trained using the multi-threshold update scheme. The accuracy is comparable to that of prior ASIC design⁵.”

8. There are several works in literature on neuromorphic CMOS silicon cochlea circuits, some even showing sound source localization. It will be helpful if the authors can also include and compare against these works their proposed technique based on memristive devices. And show where memristive circuits are bringing the true benefit.

Reply: We thank the reviewer for the valuable suggestion. Some references about CMOS-based silicon cochlea circuits have been added in the manuscript⁶⁻⁸. As presented in **Fig R7**, take a CMOS-based cochlea system as an example, RF cochlea circuits, including a transmission-line active cochlear model, are used for RF spectrum analysis. To estimate the angle of the sound source, the digitized continuous

wavelet transform (CWT)-like output vectors of two cochlea are analyzed by a digital processor. It faces the well-known obstacle that computing efficiency is gradually failing to keep up with the increasing demand. In this work, we focus on the hardware implementation of localization function with memristor arrays. Benefiting from the computing-in-memory architecture and brain-like analogue computing paradigm, memristor-based sound localization distinguishes itself from the existing ASIC work with significant energy saving⁵. In future work, it is considered to replace the signal processing module implemented by PC with memristor array to achieve a complete high-efficient system.

Fig R7 Block diagram of the experimental prototype of a two-channel RF scene analysis system based on two RF cochlea chips⁷.

Some CMOS-based silicon cochlea works have been added in the manuscript (page 2, line 11). Please see the revised manuscript. “Complementary metal-oxide-semiconductor (CMOS) circuits have been widely employed to detect the received time difference of binaural sound signals⁵⁻⁹.”

9. Please provide details of the selector device and RRAM set/reset current values.

Reply: We greatly appreciate the comments by the reviewer. In this work, one-transistor-one-memristor (1T1R) is used in the experimental demonstration of sound localization task. Under 130nm technology node, transistors are fabricated in the standard CMOS foundry¹⁰. The transistor length and width are 1 μ m and 0.5 μ m. Fig R8 shows the typical I-V curve of a memristor device (SET voltage: 0.9V, SET current 52.7 μ A; RESET voltage: -0.7V, RESET current 51.6 μ A).

Fig R8 Typical I-V curve of a memristor cell by DC sweep.

More details about the memristor device have been added (Memristor array fabrication section). “Transistors are fabricated in the standard CMOS foundry. The length and width are 1μm and 0.5 μm.” Please see the revised manuscript.

10. Can the authors comment on the impact of device endurance on the proposed methodology?

Reply: Thanks for the detailed comments by the reviewer. First, during the *in-situ* training process, with weak operation pulses, the conductance is incrementally updated. It is different from the write operation of digital memory application with full window switching. The lifetime of analogue memristor with incremental switching is much longer¹¹, up to more than 10¹¹ update number. A typical training process may only require 10⁵ update number on each weight. Therefore, the endurance lifetime is not an insurmountable bottleneck for training on analogue memristors.

Second, the total number of update pulses during the training process with our proposed method is less than the existing scheme, as shown in **Fig R9**. Therefore, compared to the traditional training scheme, our method has a higher tolerance for device endurance.

Fig R9 The total update times of memristor-based sound localization under various numbers of update thresholds

11. Can the authors comment on the impact of RRAM intermediate resistance state retention for the proposed methodology?

Reply: We thank the detailed comments by the reviewer. **Fig R10** presents the retention characteristics of 8 conductance levels of 128 memristor devices baking at 125°C.

Fig R10 Read current distribution of 8 conductance levels at 125°C¹².

As the baking time increases, the distributions of read current become wider. Different from memory application, the neural network has a certain tolerance for retention. The localization performance is simulated with analogue memristor retention model¹². As depicted in **Fig R11**, the angle deviation of sound localization network remains basically unchanged during a relative long period. 10^4 seconds under 125°C corresponds to 2.4 years under 60°C (activation energy is set to 1.58)¹². The localization accuracy of the network degrades after 10^5 seconds baking time and weight update operations are required.

Fig R11 Retention effect on angle deviation of sound localization network at 125°C.

12. Which type of resistive switching mechanism is at play in the devices the authors have used? This information should be discussed in the paper.

Reply: We greatly appreciate the suggestion by the reviewer. To achieve bidirectional analogue conductance modulation behavior, a TiN/TaO_y/HfO_x/TiN stack structure is adopted with the material proportion delicately designed. HfO_x acts as the switching layer. The TaO_y layer acts as thermal enhanced layer, whose low thermal conductivity confines the heat in the HfO_x layer, inducing the increasing temperature of conductive filament (CF) region. It makes distribution of oxygen vacancies more uniform¹³. As shown in Fig R12, when an appropriate voltage is applied to the electrodes of device, multiple weak CFs are formed and beneficial for analogue behavior.

Fig R12 Schematic diagram of two type of memristors, (a) single strong CF. (b) multiple weak CFs.

We have added some explanations about the mechanism of memristor (Memristor array fabrication section). “The 8nm-thick HfO_x acts as the switching layer. The 45nm-thick TaO_y layer acts as thermal enhanced layer, whose low thermal conductivity confines the heat in the HfO_x layer, inducing the increasing temperature of conductive filament (CF) region and more uniform distribution of oxygen

vacancies¹³. Therefore, this memristor stack provides resistive switching with good analogue characteristics.” Please see the revised version.

13. Lines 189-191: Can the authors explain how this helps to increase the accuracy?

Reply: Thanks for the detailed comment by the reviewer. For the zero-threshold-update scheme, an update pulse is given based on the sign of the calculated ΔW . However, for the sound localization task, this simplified scheme is not suitable. To illustrate, with this scheme, the distribution of weight update values of various angle samples is simulated, as shown in **Fig. 3(a)** in the main text. For some large update values, the conductance change after one SET/RESET pulse is far from reaching the target value. On the other hand, for a small update value, one pulse may change the conductance much larger than the targeted value. Therefore, it is difficult to find a universal pulse operation condition to update all weights as similar as possible to the desired values, demanding weight adaptation in subsequent cycles. For the multi-thresholds-update scheme, the calculated ΔW is compared with several thresholds and distributed into different update levels. The number of update pulses is determined based on the comparison results. The updated conductance value is closer to the target value. From the simulations and experiments, it is more suitable for the *in-situ* training of sound localization.

We have added more explanations about the multi-thresholds-update scheme (Training schemes section). “The calculated $|\Delta W|$ is sequentially compared with $|W_1|, \dots, |W_M|$. According to the intervals at which the update weights are located, the number of update pulses is determined based on the comparison results.” “The updated conductance value is closer to the target value.” Please see the revised manuscript.

Supplementary Sheet

14. Supp Fig 3: Can the authors provide equations and reference for the device model used?

Reply: Thank you very much for your detailed comments. We develop a behavior model for our analogue memristor device. Here, G is the current conductance state. The conductance between G_{\min} and G_{\max} is divided into 10 intervals. The conductance is calculated by the following equation:

$$G' = G + (a + b * \sigma_{wr}). \quad (3)$$

In this equation, a and b are both fitting parameters in different intervals, as presented in **Sup Fig. 3**. G is the current conductance state. σ_{wr} is a random number related to variance scale factor. G' is the next conductance value after applying a SET/RESET pulse.

We have added more descriptions about the memristor model, as shown in **Supplementary note 3**. Please see the revised version.

15. Supp fig 4: Can the authors provide comparison of the simulated curves and the measured curves of the real fabricated devices with both D2D + C2C variation?

Reply: We greatly appreciate the comments by the reviewer. In this model, we consider the device-to-device variation. The simulated and measured curves of memristor devices are shown in **Fig R13**.

Fig R13 (a) Simulated curves of the memristor devices when the variance scale factor equals 1.0. (b) Measured curves of the actual memristor devices.

16. Supp Fig 6: Please elaborate on what is meant by tester in the middle of the picture.

Reply: We do appreciate the reviewer for this valuable comment. The tester is the measurement equipment that connected with the PC and probe card. During the training process, according to received commands, the tester generates SET/RESET/READ pulses and sends them to a probe card that is connected to the memristor array. The current flow through SLs is collected by the probe card and sent back to the tester. Then, the tester delivers data to the connected PC for further analysis.

More descriptions about the tester have been added in the caption of **Sup Fig. 6**. “The tester is the measurement equipment that is connected with the PC and probe card. During the training process, according to received commands, the tester

generates SET/RESET/READ pulses and sends them to a probe card that is connected to the memristor array. The current flow through SLs is collected by the probe card and sent back to the tester. Then, the tester delivers data to the connected PC for further analysis.” Please see the revised version.

17. Supp Fig 8 and Tab 1: Please specify the precision of the output localization range as that is essential. The presented solution has limited precision of up to 40 degrees compared to the CMOS solution which has a resolution of 2 degrees. Can the authors comment on that aspect?

Reply: We thank the reviewer for the detailed comments. For the single-layer localization network, the average detection precision is 9.94 degree in range of -90 degree to 90 degree. Limited by scale, the performance of single-layer network is not good as the CMOS design with lots of logic gates or digital processor. To obtain a detection accuracy comparable to that of the CMOS design, a larger network (200-50-11) is considered in the simulation process. 11 output neurons contribute for the final detection results. The synaptic weights are firstly trained by software and then are mapped and programmed on the memristor arrays with deviation of $2\mu\text{S}$. After that, the final layer is online trained using the multi-threshold update scheme. As shown in Fig R14, the average angle deviation of trained network is less than 5 degrees, comparable to the existing ASIC work⁵.

Fig R14 Detection results of sound localization network with the memristor model.

To obtain a detection accuracy comparable to that of the CMOS design, evaluation results about a larger network (200-50-11) with a comparable accuracy have added in the supplementary information. “We evaluate the hardware cost of a larger memristor-based sound localization network with 200 input, 50 hidden and 11 output neurons. Output neurons contribute for the final detection results. The synaptic weights are firstly trained by software and then are mapped and programmed on the memristor arrays with deviation of $2\mu\text{S}$. After that, the final layer is online trained using the multi-threshold update scheme. The accuracy is comparable to that of prior ASIC design.” Please see the updated **Supplementary note 9**.

18. Supp Fig. 7: it is not clear which elements were fabricated in CMOS which all were simulated. Can the authors please clarify this.

Reply: Thanks for detailed comments by the reviewer. In the evaluation of sound localization, data of the BL driver, WL drivers and integrators are actual test results¹. The performance of shift adder circuit is simulated under 130nm technology node². For the soundwave pre-processing, the latency and power of the actual CMOS-based Fourier transformation module can be observed in the reference³. Measured data of flash ADC comes from the reference⁴.

In the revised manuscript, we have added descriptions about the evaluation details (Benchmarking for hardware implementation section). “Performance of the BL driver, WL drivers and integrators comes from test results of actual circuit¹. The shift adder module is simulated by XPEsim under 130nm technology node². The data of the actual Fourier transformation circuit are obtained through reference³. Measured data of flash ADC comes from the reference⁴.” Please see the revised manuscript.

19. For the RRAM device can the authors share more details like layer thickness etc.

Reply: We greatly appreciate the comments by the reviewer. For the TiN/TaO_y/HfO_x/TiN memristor, the thickness of the TaO_y and HfO_x layers are respectively 45 nm and 8nm. The device area is 0.5μm×0.5μm.

The descriptions about device fabrication details have been added in the manuscript (Memristor array fabrication section). “The 8nm-thick HfO_x acts as the switching layer. The 45nm-thick TaO_y layer acts as the thermal enhanced layer, whose low thermal conductivity confines the heat in the HfO_x layer, inducing the increasing temperature of conductive filament (CF) region and more uniform distribution of oxygen vacancies¹³. Therefore, this memristor stack provides resistive switching with good analogue characteristics.” Please see the revised version.

Minor Comments

20. Please re-verify the formatting of all references. Title is missing in some places.

Reply: We thank the detailed comments by the reviewer. We have corrected the format of references in the revised manuscript.

We have revised the format of all references. Please see the revised version.

21. fig 1c is indicating selector free crossbar?

Reply: We greatly appreciate the detailed comments by the reviewer. The **Fig. 1(c)** in the manuscript has been corrected, as shown in **Fig R15**. 1T1R architecture is used in this work. The transistor connected with a memristor cell acts as a selector and current limiter.

Fig R15 The sound localization and hardware implementation.

To avoid ambiguity, the **Fig 1c** has been revised. Please see the revised manuscript.

22. In Fig 4 some axis labels are too small to read.

Reply: We thank the reviewer to point out this issue. The Fig. 4 in the original manuscript has been replaced with the Fig R16 below.

Fig R16 Experimental demonstration results of sound localization.

The original Fig. 4 has been replaced with Fig R16. Please see the revised version.

References

- 1 Liu, Q. *et al.* A Fully Integrated Analog ReRAM Based 78.4TOPS/W Compute-In-Memory Chip with Fully Parallel MAC Computing. *2020 IEEE International Solid-State Circuits Conference - (ISSCC)*, 500-502, doi:10.1109/ISSCC19947.2020.9062953 (2020).
- 2 Zhang, W. *et al.* Design Guidelines of RRAM based Neural-Processing-Unit: A Joint Device-Circuit-Algorithm Analysis. *2019 design automation conference(DAC)* (2019).
- 3 Yuan, G. *et al.* Memristor crossbar-based ultra-efficient next-generation baseband processors. *2017 IEEE 60th International Midwest Symposium on Circuits and Systems (MWSCAS)*, 1121-1124, doi:10.1109/MWSCAS.2017.8053125 (2017).
- 4 Paulus, C. *et al.* A 4GS/s 6b flash ADC in 0.13 μm CMOS. *2004 Symposium on VLSI Circuits. Digest of Technical Papers (IEEE Cat. No.04CH37525)*, 420-423, doi:10.1109/VLSIC.2004.1346637 (2004).
- 5 Jin, J. *et al.* Real-time Sound Localization Using Generalized Cross Correlation Based on 0.13 μm CMOS Process. *JSTS:Journal of Semiconductor Technology and Science*,

- doi:10.5573/JSTS.2014.14.2.175 (2014).
- 6 Xu, Y. *et al.* A Biologically Inspired Sound Localisation System Using a Silicon Cochlea Pair. *Applied Sciences* **11**, 1519 (2021).
- 7 Wang, Y. & Mandal, S. Bio-Inspired Radio-Frequency Source Localization Based on Cochlear Cross-Correlograms. *Front Neurosci-Switz* **15**, doi:10.3389/fnins.2021.623316 (2021).
- 8 Wang, Y., Mendis, G. J., Wei-Kocsis, J., Madanayake, A. & Mandal, S. A 1.0-8.3 GHz Cochlea-Based Real-Time Spectrum Analyzer With Δ - Σ -Modulated Digital Outputs. *IEEE Transactions on Circuits and Systems I: Regular Papers* **67**, 2934-2947, doi:10.1109/TCSI.2020.2990364 (2020).
- 9 Nguyen, D., Aarabi, P. & Sheikholeslami, A. Real-Time Sound Localization Using Field-Programmable Gate Arrays. doi:10.1109/ICME.2003.1221745 (2002).
- 10 Yao, P. *et al.* Fully hardware-implemented memristor convolutional neural network. *Nature* **577**, 641-646, doi:10.1038/s41586-020-1942-4 (2020).
- 11 Zhao, M. *et al.* Characterizing Endurance Degradation of Incremental Switching in Analog RRAM for Neuromorphic Systems. *2018 IEEE International Electron Devices Meeting (IEDM)*, 20.22.21-20.22.24, doi:10.1109/IEDM.2018.8614664 (2018).
- 12 Zhao, M. *et al.* Investigation of statistical retention of filamentary analog RRAM for neuromorphic computing. *2017 IEEE International Electron Devices Meeting (IEDM)*, 39.34.31-39.34.34, doi:10.1109/IEDM.2017.8268522 (2017).
- 13 Wu, W. *et al.* A Methodology to Improve Linearity of Analog RRAM for Neuromorphic Computing. *2018 IEEE Symposium on VLSI Technology*, 103-104, doi:10.1109/VLSIT.2018.8510690 (2018).

Reviewer #2:

1. It is not clear what is the key advantage of using memristor based VMM for this particular sound localization task. The authors claim that the key challenge of this application is to have a "vote" at the output layer, but this seems to me to be what you are doing if you consider the same classification task with a 2 layer (or more) neural net. Also, the challenge of having multiple output neurons to optimized (and not only one) is only due to the fact that the authors are using a small 1 layer network. This is what is expected when considering the difference between classification and pattern matching.

Reply: We are sorry that the reviewer misunderstood the motivation of this work. In this work, we develop a memristor based brain-like hardware for sound localization task. Localization is one of the most common AI tasks. The method proposed in this work is not only applicable to a particular sound localization task, but also other typical localization tasks. Previous demonstrations on localization tasks were based on digital CMOS circuits and von-Neumann architecture. This work takes the advantage of brain-like behaviors of memristor devices, including the computation-in-memory and analogue computing capability, to enhance the energy efficiency and get the *in-situ* training ability.

In the sound localization, a representative analogue computing application, the major challenge is that the input and output are both continuous values, instead of a digital integrating process. In the prior work, as the reviewer mentioned, the sound localization application is converted into classification tasks¹. In the fully-connected layer, each output neuron corresponds to a certain direction label and refers to a discrete angle value. The maximum output neuron represents the predicted angle range instead of providing a definite prediction value. To obtain a high precision localization result, a lot of output nodes are desired, posing a great challenge to hardware overhead and energy consumption. In contrast, the all-analogue computing method benefits from the significant reduction of the network scale, the hardware cost is much smaller than a classification network.

However, the all-analogue computing tends to be susceptible to errors, especially during *in-situ* training process. The error accumulation may cause significant accuracy loss. So, in this work, we solve this problem by using optimized memristor device and a proposed multi-threshold weight update scheme. We realize *in-situ* training on the memristor array and get good accuracy. Our method is applicable to both single-layer network and multi-layer network. In the revised manuscript, we also simulate an all-analogue multi-layer network, demonstrating the scalability of our proposed method.

We have added some explanations about the sound localization network (page 3, line 1 and line 9). “Typically, in a classification application, each output neuron corresponds to a certain label and refers to a discrete value. The main target is to maximize the ‘right’ one. The neural networks used for these tasks are redundant². As a result, in many complex cases, a lot of output nodes are desired, posing a great challenge to hardware overhead and energy consumption.” “Benefits from the significant reduction of the network scale, the hardware cost is much smaller than a classification network. However, since every synapse and neuron influence the output result, this analogue computing task tends to be susceptible to errors. In addition, for training samples corresponding to various directions, the supervised values of the target neurons are different. Therefore, in the feed-forward and training process, this task is less tolerant of inevitable non-ideality of hardware, putting forward higher requirements for device performance and weight update approach.”

Besides, we also have added simulation results of two-layer sound localization network with multi-thresholds-update scheme (page 8, line 13). “This method also applicable for more complex tasks, as shown in **Sup Fig. 6** and **Sup Fig. 7**. Depending on the device characteristics and tasks, the thresholds and pulse number may be different in other works.” Please see the revised manuscript.

2. The second major claim is to have a 47% improvement of performances with 177% reduction in energy consumption. OK, but could you compare the improvement in energy consumption for the same performances? It seems to me that 46% is for the particular hardware of the study (i.e. different threshold methods) and 177% is in between different hardware.

Reply: We thank very much for your detailed comments. We may not explain the data mapping schemes clearly in our previous manuscript. With the conventional without verification scheme, the memristor based network after training shows a large deviation for sound localization task. With the proposed multi-thresholds-update scheme, the *in-situ* training result of sound localization is improved by 45.7% compared to the existing scheme, while the energy consumption is also reduced. With different numbers of thresholds, the comparison results of energy consumption and processing time cost on memristor array are illustrated in **Fig 3e** and **Sup Fig. 5** in the manuscript and supplementary information.

In addition, we evaluate the hardware cost of a larger memristor-based sound localization network. The accuracy is comparable to that of prior CMOS ASIC design³. The updated comparison table is presented in **Table R3**. Compared to the CMOS design, the memristor-based sound localization yields improvement of 3571× in the processing time, and 184× in the energy cost.

Design	Technology node/nm	Processing time/ms	Energy consumption/ μJ
Memristor-based	130	0.0028	0.306
ASIC ³	130	10	56.3
Improvement to ASIC	-	3571 \times	184 \times

Table R3 The final evaluation results for memristor-based sound localization and comparisons with the ASIC design³.

The evaluation results have been updated, as illustrated in **Supplementary note 9**. “We evaluate the hardware cost of a larger memristor-based sound localization network with 200 input, 50 hidden and 11 output neurons. Output neurons contribute for the final detection results. The synaptic weights are firstly trained by software and then are mapped and programmed on the memristor arrays with deviation of $2\mu\text{S}$. After that, the final layer is online trained using the multi-threshold update scheme. The accuracy is comparable to that of prior ASIC design³.” Please see the revised version.

3. The most interesting part of the paper is the threshold method used to circumvent the nonlinear weight update and low accuracy of memristor tuning. This aspect would deserve a more complete analysis such as accuracy improvement with respect to other technics, linearity of the weight updates, mapping between threshold and number of pulses for update, etc...

Reply: Thanks for valuable comments by the reviewer. We investigate and simulate the *in-situ* training performance with nonlinearity characteristic as well as various update schemes. Detailed results and discussions are shown below:

1) Weight update schemes

In-situ with verification and without verification scheme, are commonly used in the previous training works⁴. For the former update scheme, we simulate the detection deviation of sound localization network with various programming errors. As presented in **Fig R17**, the localization accuracy degrades slightly as the programming error increases. Compared to the *in-situ* with verification scheme, the proposed scheme in this work could also achieve an acceptable accuracy without complicated verification circuit⁵⁻⁷ and at least one order of magnitude more programming pulses.

Fig R17 Simulated effect of different programming errors on the detection deviation for the *in-situ* with verification scheme. The green line reflects the actual experimental result of two-thresholds-update scheme.

In-situ without verification scheme is also a common solution in previous study. It is referred to the zero-threshold-update scheme in this work. As shown in **Fig. 3(d)**, the proposed two-thresholds-update scheme yields improvements of 4~5 degrees in the training performance, relative to the traditional without verification scheme. The processing time and energy consumption are respectively reduced by 9% and 11%.

Considering the hardware cost and training accuracy, the two-thresholds-update scheme is the most suitable solution for the sound localization task in contrast to common two solutions.

2) Nonlinearity

As illustrated in **Fig R18**, we simulate the training accuracy with linear and nonlinear device models. For different nonlinearity characteristics, different training schemes are adopted. The thresholds and pulse numbers are the best solutions currently explored for various models. As depicted as **Fig R18(b)**, it is worth noting that with the introduction of one update threshold, the nonlinear device shows a better training result compared to the linear device. For the nonlinear analogue behavior, with the low initial state, the increase of conductance is greater than that of the linear model. As presented in **Fig R19**, it exhibits a wider range of conductance change, making a higher training accuracy.

When the number of thresholds is greater than 2, the training results of sound localization network will tend to be saturated. The linear model shows a slightly better training effect than the nonlinear model.

Fig R18 (a) The device models with different nonlinearity. Curves 1,2,3 respectively show device models with increasing nonlinearity values. Curve 2 is closer to the actual device. (b) Simulated training accuracy with multi-thresholds-update scheme. Different color lines represent nonlinear and linear device models in the (a).

Fig R19 Conductance change of 100 cells in first weight array. (a) (b) (c) respectively correspond to nonlinearity value 1,2,3 with the one-threshold-update scheme.

3) Update thresholds

Fig R20 presents the schematic diagram of multi-thresholds-update scheme. As the number of updating levels increases, there will be more thresholds for ΔW classification to apply different numbers of pulses on the memristor device. The thresholds and pulse numbers in multi-thresholds-update scheme, are determined by simulation.

Take the parameters of the two-thresholds-update scheme as an example, the two thresholds W_1, W_2 determines range of update weights. Fig R21 shows the *in-situ* training accuracy with the various W_1 and W_2 . It can be illustrated that to obtain a satisfactory result, W_1 and W_2 values are required to be in the appropriate range. The scheme ($W_1=1\mu\text{S}$, $W_2=10\mu\text{S}$) used in the experiment is the best solution currently explored. Through more exploration, the training effect may be further improved slightly in the future.

Fig R20 Schematic diagram of the multi-threshold-update scheme.

Fig R21 *In-situ* training results with different W_1 and W_2 values.

4) Pulse number

In the two-thresholds-update scheme, the $|\Delta W|$ in different ranges correspond to the programming pulse number used in that case ($n_0 < n_1 < n_5$, n_0 and n_1 are generally 0,1). Fig R22 presents the impact of different n_5 on the training accuracy. Increasing n_5 within a certain range is conducive to improving the training result. However, when n_5 is greater than 150, the continued increase pulse number will cause a larger deviation between target and actual ΔW , requiring subsequent update operations for compensation. In this case, the training effect is worse.

Fig R22 *In-situ* training results with the various n_s values.

More analysis about the device characteristics has been added in the revised **Supplementary note 8**. Please see the revised supporting information and **Sup Fig. 8** and **Sup Fig. 9**. “The impact of asymmetric behavior on the performance of memristor-based sound localization. (a) The device model with asymmetric and symmetric behaviors. (b) Training accuracy with various number of thresholds. (c) Processing time spent on the memristor array during *in-situ* training process. (d) Energy consumption spent on the memristor array during *in-situ* training process”; “The impact of nonlinearity behavior on the performance of memristor-based sound localization. (a) The device models with different behaviors. Curves 1,2,3 respectively reflect device models with increasing nonlinearity values. Model 2 is closer to the actual device. (b) Simulated training accuracy with multi-thresholds-update scheme. (c) Processing time spent on the memristor array under various number of update thresholds. (d) Energy consumption spent on the memristor array under various number of update thresholds and device models.”

4. - why low accuracy memristor can implement *in-situ* training while it is known that resolution during *in-situ* training is a strong limitation for application: is it only task related or is it thanks to the threshold method?

Reply: We greatly appreciate the comments by the reviewer. For the analogue memristor, the conductance is capable of continuously adjusting with SET/RESET programming pulses. Compared to digital devices, this analogue behavior of memristor is similar to the biological brain and more suitable for *in-situ* training process. Many previous works have experimentally demonstrated *in-situ* training with memristors on classification tasks, indicating the resolution of analogue memristor is enough for *in-situ* training.

However, the localization tasks are more difficult than classification tasks. The device variations and reliability issues (e.g. state-stuck, conductance drift, etc.) may decrease the training accuracy significantly. Therefore, to our best knowledge, there has been few works that demonstrated all-analogue localization task with the

memristor array. In this work, by using optimized memristor devices and the proposed multi-thresholds weight update method, we realize good training accuracy on the sound localization. The proposed multi-thresholds weight update method is to suppress the influence of device variations. So, this method is not task related, but can be used on different tasks including classification tasks. For more complex tasks, the multi-thresholds-update scheme is also applicable, but more thresholds may be required.

We have added descriptions about the proposed *in-situ* training scheme (page 8, line 13). “This method also applicable for more complex tasks, as shown in **Sup Fig. 6** and **Sup Fig. 7**. Depending on the device characteristics and tasks, the thresholds and pulse number may be different in other works.” Please see the revised version.

References

- 1 Jiang, S., Wu, L., Yuan, P., Sun, Y. & Liu, H. Deep and CNN fusion method for binaural sound source localisation. *Journal of Engineering-Joe* **2020**, 511-516, doi:10.1049/joe.2019.1207 (2020).
- 2 Yu, S. M. *et al.* Binary Neural Network with 16 Mb RRAM Macro Chip for Classification and Online Training. *Int El Devices Meet* (2016).
- 3 Jin, J. *et al.* Real-time Sound Localization Using Generalized Cross Correlation Based on 0.13 μm CMOS Process. *JSTS:Journal of Semiconductor Technology and Science*, doi:10.5573/JSTS.2014.14.2.175 (2014).
- 4 Yao, P. *et al.* Face classification using electronic synapses. *Nat Commun* **8**, doi:10.1038/ncomms15199 (2017).
- 5 Zangeneh, M. & Joshi, A. Design and Optimization of Nonvolatile Multibit 1T1R Resistive RAM. *Ieee T Vlsi Syst* **22**, 1815-1828, doi:10.1109/Tvlsi.2013.2277715 (2014).
- 6 Garcia-Redondo, F. & Lopez-Vallejo, M. Self-controlled multilevel writing architecture for fast training in neuromorphic RRAM applications. *Nanotechnology* **29**, doi:10.1088/1361-6528/aad2fa (2018).
- 7 Chen, J. *et al.* Optimization Strategy for Accelerating Multi-Bit Resistive Weight Programming on the RRAM Array. *2019 IEEE International Workshop on Future Computing (IWOFC)*, 1-3, doi:10.1109/IWOFC48002.2019.9078447 (2019).

Reviewer #3:

Summary: In this paper, a scheme is presented for training a memristor-based sound localization network in situ that uses thresholds of ΔG to determine how many SET or RESET pulses should be applied to a memristor weight. This helps compensate for the stochastic change in conductance for a given initial conductance, which reduces the likelihood that the change in conductance will be too large or too small. This scheme is compared to previous schemes, where the change in conductance is either verified (requires more energy to double-check) or left unverified (less accurate).

1. Strengths: The paper is technically sound, describing the memristor-based network operation in detail and supporting results with plotted data. The driver, ADC, inverter, etc. circuits are considered when estimating the power. A 45.7% improvement in accuracy and a 177x decrease in energy consumption are reported, which are credible and impressive results.

Reply: Thanks for recognizing the importance of our work and the proposed brain-like memristor-based sound localization.

2. Impact: As previously mentioned, I believe that in situ training is an important topic in analog computing because of the nonvolatile nature and interesting dynamics of memristors. Sound localization is just one of many potentially useful applications. However, this work seems to focus on updating memristor values accurately and efficiently. This goal is important, though, with such emphasis on in situ training, some novelty in the training method (how is the ΔG calculated?) might further improve the impact of this work.

Reply: We greatly appreciate the comments by the reviewer. Based on the reviewer's comment, the statement of *in-situ* training in the introduction section have been modified.

Regarding the training method, more descriptions about the calculation of ΔW have been added in the manuscript. For the single-layer sound localization, the gradient descent method is employed to calculate update weight matrix $\Delta W_{i,j}$, as shown in the following equation:

$$\Delta W_{i,j} = \eta * x_i * (y_j - \hat{y}_j) * \frac{\partial y_j(x_i)}{\partial x_i} \quad (4)$$

In the training process, η and x are respectively the learning rate and inputs. y and \hat{y} refer to the actual and target outputs. The minibatch is selected as 5. Therefore, in

each update iteration, the update weight is calculated as the average value of 5 feedforward calculation results.

We have revised the introduction part of manuscript (page 3, line 14). “Therefore, in the feed-forward and training process, this task is less tolerant of inevitable non-ideality of hardware, putting forward higher requirements for device performance and weight update approach.” Regarding the calculation of ΔW , we have added more description in the Sound localization algorithm section. “In the training process, η and x are respectively the learning rate and inputs. y and \hat{y} refer to the actual and target outputs. In the experiments, minibatch is chosen as 5. Therefore, in each update iteration, the update weight is calculated as the average value of 5 feedforward calculation results.” Please see the revised manuscript.

3. Thoughts: This work produced promising results (45.7% improvement in accuracy and 177x decrease in energy consumption) in the important field of in situ training. I also feel that while the work is technically sound and novel, it does not seem particularly technically difficult. I think that if the focus is on in situ training specifically instead of just efficient/accurate memristor programming, work should also perhaps be done in terms of calculating gradient or implementing backpropagation in general. In conclusion, I would accept this paper, but with slight modifications to the motivation section.

Reply: We sincerely thank the reviewer’s effort and valuable suggestions to help us improve our manuscript. In the revised version, some statements about the motivation have been added. The accuracy with widely used *ex-situ* training is easily affected owing to variations and reliability issues. The practical *in-situ* training, could adapt to the inevitable environment change and non-ideal parameters of hardware (for example, programming variations, state-stuck devices, conductance drift, and so on)^{1,2}. According to the real-time output deviation, the weight values are adjusted. To get high training accuracy on the localization task, optimized memristors with higher performance and an improved weight update scheme during *in-situ* training process are required. This is the key motivation of our work.

In the revised version, relative descriptions about motivation have been corrected (page 6, line 15; page 3, line 14). “The *ex-situ* training is easily affected by non-ideal parameters of hardware, for example, programming variations, state-stuck devices, conductance drift, and so on^{1,2}. By adjusting weight values according to the real-time output deviation, the *in-situ* training method² could adapt well to environmental changes”. “In the feed-forward and training process, this task is less tolerant of inevitable non-ideality of hardware, putting forward higher requirements for device performance and weight update approach.” Please see the revised version.

References

- 1 Wang, Z. *et al.* In situ training of feed-forward and recurrent convolutional memristor networks. *Nature Machine Intelligence* **1**, 434-442, doi:10.1038/s42256-019-0089-1 (2019).
- 2 Alibart, F., Zamanidoost, E. & Strukov, D. B. Pattern classification by memristive crossbar circuits using ex situ and in situ training. *Nat Commun* **4**, 2072, doi:10.1038/ncomms3072 (2013).

Reviewers' Comments:

Reviewer #1:

None

Reviewer #2:

Remarks to the Author:

The authors provided a substantial review of the paper. The article is clear and scientifically sound. I still maintain my major criticism, which is about the impact of the paper. The claim of performances is not fair with what is done in the community. For example, ref3 from the response letter is showing 92.5% performances on this task with deviation angle of approximately 5° . This result seems to me quantitatively better than what is reported by the authors (8 to 10° deviation, and no mention of total performances on the task). The paper should mention explicitly what are the absolute performances with respect to SotA. The discussion should establish the trade off in between energy improvement, speed AND accuracy. The low performances of memristor-based approaches is for the moment not solved and low accuracy weight result in all reported work (in situ and ex situ) in a decrease of performances. This is an essential research question that needs to be mentioned explicitly.

In my opinion, the main contribution of the paper is still the multi threshold approach, which is interesting for improving memristor-based hardware, but which is not yet capable of bringing memristor hardware at the level of existing solutions.

Reviewer #3:

Remarks to the Author:

The revised version has well addressed all my queries, especially in the motivation section. I don't have more questions for the authors.

Reply to Reviewers' Comments:

Reviewer #2:

1. The authors provided a substantial review of the paper. The article is clear and scientifically sound. I still maintain my major criticism, which is about the impact of the paper. The claim of performances is not fair with what is done in the community. For example, ref3 from the response letter is showing 92.5% performances on this task with deviation angle of approximately 5° . This result seems to me quantitatively better than what is reported by the authors (8 to 10° deviation, and no mention of total performances on the task). The paper should mention explicitly what are the absolute performances with respect to SotA. The discussion should establish the trade off in between energy improvement, speed and accuracy. The low performances of memristor-based approaches is for the moment not solved and low accuracy weight result in all reported work (in situ and ex situ) in a decrease of performances. This is an essential research question that needs to be mention explicitly. In my opinion, the main contribution of the paper is still the multi threshold approach, which is interesting for improving memristor-based hardware, but which is not yet capable of bringing memristor hardware at the level of existing solutions.

Reply: We sincerely thank the reviewer's effort and valuable suggestions to help us improve our manuscript. Below is our point-to-point response to your concerns.

Accuracy improvement with a larger network: As the reviewer pointed out, the deviation angle in our experimental demonstration is larger than previous CMOS based digital circuits. This is mainly due to the limited network size of our demonstration. In this work, we use a 128×8 array to demonstrate a single-layer network. This network size is too small to get a high localization accuracy as a large-scale CMOS circuit does. Since memristor is not a mature technology like CMOS, most of the present demonstrations are based on single small array. In the future work, as technology grows mature and the array size increases, the localization accuracy can be much improved.

To obtain a better detection accuracy, a larger network (200-50-11) is simulated based on our device model. The model description has been added in the supplementary information according to the reviewer's comments in the 1st round of review. In summary, our device model considers all the non-ideal effects of our memristor device, including the conductance variations, nonlinearity, weight update precision, and so on. The model parameters are all extracted based on the measurement results. So the model can well capture the real device's behaviors.

In this two-layer network simulation, as depicted in **Table R1**, the root mean square localization error (RMSE) of the azimuth at each position is about 5.7 degrees for binaural signals. In general, the localization accuracy of the memristor-based network is comparable to the reported CMOS systems for the similar task. Noted that the RMSE of our memristor-based network is still 1.5 degrees larger than the CMOS ASIC work¹ which was mentioned by the reviewer. There are two reasons: one is due to the non-ideal effect of the memristor devices, the other is that the CMOS ASIC work used more signal sources while our work only used two. Therefore, there is still much room to improve the accuracy for the memristor-based network in future.

Design	Hardware learning	RMSE (degrees)
Memristor-based	YES	12.5 (single-layer network, experiment) / 5.7 (two-layer network, simulation)
DSL system¹	NO	4.2
Three-chip CMOS system²	NO	5
MEMS system³	NO	8.7

Table R1 The RMSE results for single-layer and two-layer memristor network and comparisons with other designs, including dedicated sound localization circuit (DSL) system¹, a three-chip CMOS system², and a MEMS system³.

Performance benchmark: To make a relative fair comparison, we use the 200-50-11 network for the performance benchmark. Under the same technology node, the two-layer memristor-based sound localization network yields improvement of 3571× in the processing speed, and 184× in the energy efficiency (**Table R2**). Compared with the existing ASIC work with conventional computation architecture and digital circuits¹, with a small degradation in the localization accuracy, the memristor-based system has greater advantages in terms of energy efficiency and processing speed.

Design	Technology node /nm	Processing time /ms	Energy consumption /μJ
Memristor-based	130	0.0028	0.306
ASIC¹	130	10	56.3
Improvement factor	-	3571×	184×

Table R2 The benchmark results for memristor-based sound localization and comparisons with the sound localization chip (ASIC) in DSL system¹.

Limitation of this work and future outlook: We agree with the reviewer that the reduced accuracy of memristor-based approaches is the most critical challenge in this field. Optimizing memristor's precision or system accuracy is a very hard task. A lot of studies are working toward this goal, but it still needs a long way to go. On the other hand, since memristor is a neuromorphic device, that behaves similar with biological neurons or synapses, it has great potential to construct a brain-like cognitive system. The memristor based computation-in-memory architecture shows great advantages in computing power and energy efficiency compared to the conventional architecture. Therefore, although the memristor-based systems cannot completely compete with digital approaches in accuracy at the current stage, it is still a very attractive technology. Meanwhile, some mixed architecture can be designed, in which memristor-based computation-in-memory unit works at the front-end for the fast localization, and digital unit works at the back-end for the fine calculation.

Based on the reviewer's comment, we add some discussion on the trade off in between energy improvement, speed and accuracy in the revised manuscript. We also add more discussion on the future outlook in the main text.

Main contribution of this paper: We are glad that the reviewer approves our contribution on the multi-threshold training approach. *In situ* training is an important research topic for neuromorphic computing. Previous studies mainly focused on the training for classification tasks, which are much easier than the localization tasks. In our work, we proposed the multi-threshold training approach, and then demonstrated *in situ* training for localization task on the array level. This is an important step for the memristor to process more complex neuromorphic computing tasks with high efficiency.

Revisions: We revise the evaluation and comparison results about the two-layer network. We also add more discussion about the contribution of this work, the performance trade off, and the future outlook. Detail revisions include:

Main text, page 3, line 22-line 29:

“capable of handling complete sound signals received by two artificial ears, as shown in Fig. 1b-c. With the integrated 1K HfO_x-based analogue memristor array, a subset of Head Related Transformation Function (HRTF) dataset^{38,39} is in-situ trained based on the neural network architecture. The brain-like learning algorithm of the sound localization function is successfully realized on the memristor array with a proposed in-situ training method, namely, a multi-threshold-update scheme. The tradeoff between training results and hardware overhead with different training schemes and different hardware platforms is further analyzed.”

Main text, page 9, line 6-line 20:

“Furthermore, we evaluate the hardware performance of a larger memristor-based sound localization task and compare it with previous works.” “The network has two layers with the sizes of 200×50 and 50×11 respectively.” “The evaluation details can be found in the benchmarking for hardware implementation section and Supplementary Table 2 and 3. Benefitting from computing-in-memory, with small accuracy degradation (<1.5 degrees), the memristor-based sound localization reduces the energy consumption by ~184x compared to the existing application specific integrated circuit (ASIC) designs with conventional architecture⁶. In the future work, by optimizing the weight-update nonlinearity, variations and other non-ideal effects of memristors, the training accuracy can be further improved.”

Supplementary note 8:

Design	Hardware learning	RMSE (degrees)
Memristor-based	YES	12.5 (single-layer network, experiment) / 5.7 (two-layer network, simulation)
DSL system ¹	NO	4.2
Three-chip CMOS system ²	NO	5
MEMS system ³	NO	8.7

Supplementary Table 2 The root mean square localization error (RMSE) results for single-layer and two-layer memristor network and comparisons with other designs, including dedicated sound localization circuit (DSL) system¹, a three-chip CMOS system², and a MEMS system³.

“As illustrated in Supplementary Table 2, the accuracy is comparable to the reported CMOS systems for the similar task¹⁻³.” “With small accuracy degradation, the memristor-based design respectively yields improvements of 184× in the energy cost, and 3571× in processing time compared to the previous CMOS design. This work paves the way towards highly efficient all-analogue computing task.”

References

- 1 Jin, J. *et al.* Real-time Sound Localization Using Generalized Cross Correlation Based on 0.13 μm CMOS Process. *JSTS:Journal of Semiconductor Technology and Science*, doi:10.5573/JSTS.2014.14.2.175 (2014).
- 2 Grech, I. *et al.* Analog CMOS Chipset for a 2-D Sound Localization System. *Analog Integrated Circuits and Signal Processing* **41**, 167-184, doi:10.1023/B:ALOG.0000041634.92147.0d (2004).
- 3 Van Schaik, A. & Shamma, S. A Neuromorphic Sound Localizer for a Smart MEMS System.

Analog Integrated Circuits and Signal Processing **39**, 267-273,
doi:10.1023/B:ALOG.0000029662.37528.c7 (2004).

Reviewer #3:

1. The revised version has well addressed all my queries, especially in the motivation section. I don't have more questions for the authors.

Reply: We greatly appreciate for the valuable time you have spent reviewing our manuscript and providing insightful comments to help significantly improve the quality of our work. We are very glad to see that you are satisfied with our revision.

Reviewers' Comments:

Reviewer #2:

Remarks to the Author:

The authors addressed all my points. I don't have any further comments to provide.

Dear editor,

We are glad that our manuscript “Memristor-Based Analogue Computing for Brain-Inspired Sound Localization with in situ training” has been accepted in principle. We sincerely appreciate the valuable time and effort that you and the reviewers dedicated to providing insightful feedback, which definitely improve the quality of our work.

To fully address the remaining concerns, according to the Author Checklist, we have revised the manuscript, supplementary information, figures. The authors, affiliation and acknowledgement information are double-checked. Besides, per your request, we have uploaded the source data files for all the figures in the manuscript. Please kindly find the attached revised documents.

Yours sincerely,

Huaqiang Wu, Ph.D.

Address: School of Integrated Circuits (SIC), Tsinghua University, Beijing, China, 100084

Ph: +86-10-62798608

Fax: +86-10-62771130

E-mail: wuhq@tsinghua.edu.cn

Reply to Reviewers' Comments:

Reviewer #2:

1. The authors addressed all my points. I don't have any further comments to provide.

Reply: We are grateful for the valuable comments to help significantly improve the quality of our work. We are very glad that the reviewer is satisfied with our revision.